# The Role of Biochar Nanoparticles Performing as Nanocarriers for Fertilizers on the Growth Promotion of Chinese Cabbage (*Brassica rapa* (Pekinensis Group))

Ruiping Yang [1,2], Jiamin Shen [1], Yuhan Zhang [1], Lin Jiang [1], Xiaoping Sun [1], Zhengyang Wang [3], Boping Tang [1,*] and Yu Shen [2,*]

1   Jiangsu Key Laboratory for Bioresources of Saline Soils, Jiangsu Synthetic Innovation Center for Coastal Bio-Agriculture, School of Wetlands, Yancheng Teachers University, Yancheng 224007, China
2   Co-Innovation Center for the Sustainable Forestry in Southern China, College of Biology and the Environment, Nanjing Forestry University, Nanjing 210037, China
3   Department of Environmental Sciences, The Connecticut Agricultural Experiment Station, New Haven, CT 06504, USA
*   Correspondence: boptang@163.com (B.T.); sheyttmax@hotmail.com or yushen@njfu.edu.cn (Y.S.)

**Abstract:** Chinese cabbage (*Brassica rapa*) belongs to the Pekinensis Group and is grown annually as a salad crop. It is one of the most important food crops in Eastern Asia and the most widely grown vegetable in China, accounting for more one-quarter of the total annual vegetable consumption in northern parts of the country. It is reported that nitrogen (N), phosphorus (P), and potassium (K) fertilizations play important roles in the physio-morphological traits and yields of Chinese cabbage. However, N, P, and K use in agriculture continues to increase. Excessive application of fertilizers has a harmful impact on the environment. Yet how to improve the irrigation effects on Chinese cabbage growth is still limited. In this study, we firstly selected biochar nanoparticles (BNPs) prepared from corn straw, which had been air-dried and heated in a muffle furnace at 350 °C for 120 min, with K (potassium sulfate), N (calcium nitrate tetrahydrate), and P (sodium dihydrogen phosphate dihydrate) fertilizers. Then, a screening experiment (Experiment I) was performed via the response model to find the best solution for Chinese cabbage growth. Treatment with 2 g/kg of N and 2 g/kg of K for 4 weeks was the optimum application to promote Chinese cabbage growth. Then, a comparison experiment (Experiment II) was carried out to test the best formula for Chinese cabbage growth with or without BNPs. After co-irrigation with N and K for 4 weeks, treatment with a combination of 2 g/kg of BNPs, 2 g/kg of N, and 2 g/kg of K was the optimum formula for Chinese cabbage growth. Plant biomass increased by more than 1796.86% and 32.80%, respectively, in two combined treatments of BNPs and fertilizers as compared to the control treatment. After the addition of BNPs, Chinese cabbage height (aboveground) and the dry weight of belowground biomass in the N + K treatment increased to 10.97% and 20.48%, respectively. These results suggest that BNPs have great potential as a nanocarrier for fertilization as they are highly efficient (over 50% increase), reducing fertilizer use while promoting plant growth. The use of BNPs as a nanocarrier for fertilizers represents a step toward more environmentally friendly agriculture.

**Keywords:** biochar nanoparticle; Chinese cabbage; conventional fertilizers; plant growth; nanocarrier

## 1. Introduction

Chinese cabbage (*Brassica rapa*) is known as napa, napa cabbage, petsai, wongbok, and chihli in Asian countries; and it is also called Chinese leaves or celery cabbage, which belongs to the Pekinensis group. The vegetable has a long history in China and is of major importance, with over 300,000 ha grown in China. Chinese cabbage is an important food in Korea, Taiwan, and Japan, where it is grown as an annual crop. Most Chinese cabbage cultivars are biennial and produce tight, compact, cylindrical heads [1]. Similar to other

cruciferous vegetables, Chinese cabbage has a shallow root system, which limits its ability to absorb water and nutrients from deeper soil [2]. The plant's nutritional demands are significantly higher during the growing period when the leaf mass is at its highest [3].

Chinese cabbage is a fast-growing vegetable with high nutritional value. For optimum growth, Chinese cabbage requires an adequate supply of both soil nutrients and soil water [4]. In recent decades, crop production has depended largely on the use of chemical fertilizers, with nutrient fertilization playing an important role in improving crop productivity and maintaining soil fertility. Farmers use large quantities of chemical fertilizers to increase yields. The average cost of fertilizers is a minimum of USD 23.6 per $m^2$ and 0.4 kg per $m^2$ of co-fertilizers (organic fertilizers and N-P-K fertilizers). Over 50–70% of fertilizers applied to crops/fields are not absorbed by plants but lost to surface runoff [5]. This surface runoff leads to groundwater pollution [6]. Thus, new methods are needed to improve the effectiveness of fertilizers and reduce loss lost to runoff.

In terms of nutrition, plants require a correct proportion of nitrogen (N), phosphorus (P), and potassium (K), which have a synergistic effect on plant health, plant growth, and final plant yield [7]. Fertilizers are fundamental in the development of plants and crops. Fertilizers help plants grow faster. This goal can be achieved in two ways. The first is through the use of nutrient-rich additives. The second mechanism by which certain fertilizers work is to improve the soil's efficacy by altering water retention and aeration [8]. However, it is known that fertilizers are one of the main pollution sources of soil and water, with nitrates leaking into groundwater and soil, and N and P runoff into water bodies [9]. The use of large quantities of fertilizers also contributes to greenhouse gas emissions, leading to soil pollution, bioaccumulation of pollutants in the soil, and transfer to the food chain [10,11]. Thus, how to reduce fertilizer use is important for environmental protection and sustainable agricultural development.

According to a previous study, 40–70%, 80–90%, and 50–90% of N, P, and K, respectively, applied to the soil is lost, representing a considerable cost in terms of key macronutrient resources [12]. Previous research suggested that nanocarrier-bound fertilizers exhibit higher delivery efficiencies than fertilizers applied using traditional irrigation [13]. Carbon nanomaterials, including nanotubes and graphene oxide, have been proposed as nanocarriers for micronutrients [14,15]. Although promising, thus far, most proposed nanotechnologies for micronutrient delivery have been tested only in the laboratory.

There is much interest in agricultural research in the idea of nanocarriers for fertilizers [16,17]. However, the problem of how to improve the delivery efficiency of nano-macronutrient elements remains [18]. In this respect, the current best strategy is to find a kind of material to slow nutrient release. Biochar pores are divided into micropores (<2 nm), mesopores (2–50 nm), and macropores (>50 nm), according to pore size [19]. Larger biochar pores, such as those of BNPs, offer a significantly greater specific surface area and pore volume than smaller biochar pores [20]. Biochar is known for its high potential absorption of heavy metals [21] and organic contaminants [22,23]. Nanosized materials have markedly high surface areas. Previous research proposed that biochar NPs (BNPs) have great potential as carriers for fertilizers and that they can slow down the release of fertilizers in the soil. Zein-based NPs have been shown to be a safe, biocompatible, and effective nanocarriers for botanical pest repellents [24]. Mesoporous silica nanoparticles were delivered through soil-free nutrient media to wheat translocated from roots to shoots and localized to chloroplasts, which promoted photosynthesis and seedling growth [25]. Compared with chemical fertilizers used in agriculture, BNPs are more cost effective, eco-friendly, nontoxic, and stable. In this study, it is hypothesized that BNPs are a kind of material that could reduce fertilizer use for plant growth. BNP-made nanocarriers would be effective materials for fertilizer use in sustainable agriculture in the future.

## 2. Material and Methods

*2.1. BNPs Preparation*

Fresh corn stalks were collected from Jiangning District, Nanjing, Jiangsu Province, China, and aid-dried. The waste biomass materials were then cut into 30–50 mm pieces and heated in a muffle furnace at 350 °C for 120 min. The biochar preparation method followed that of Shen et al. [26]. The aim was to convert 35% of the biomass into biochar.

After preparation of the biochar, a Planetary Ball Mill (Shunchi Tech, PMQW2; Nanjing, China) was used to process the BNPs. Ethanol was used as a grinding aid, and the weight ratio of $ZrO_2$ balls to powder was 15:1 in the vials in the system. The vials were spun at 400 rpm for 6 h. The biochar was then removed and placed in sealed bags and stored in desiccators until use. Characterization of the prepared nano biochar using transmission electron microscopy (TEM) was conducted, and X-ray diffractometer (XRD) and X-ray photoelectron spectroscopy (XPS) were performed via Da 8 Venture Single Crystal X-ray Diffractometer (Bruker, Germany), Ultima IV X-ray diffractometer (Rigaku, Japan), and Surface Area and Micropore Size Analyzer (V-Sorb 2800P, Gold APP Instruments Corporation, Xi'an, China), respectively.

In the experimental treatments, the BNPs were added to the N-P-K solution in a 1:1 (*w/w*) ratio. Before the root application, the BNPs and N-P-K solution were mixed together and then placed in a shaker at 140 rpm for 12 h. In terms of the fertilizer additions, N was derived from calcium nitrate tetrahydrate, P was derived from sodium dihydrogen phosphate dihydrate, and K was derived from potassium sulfate. All the chemicals were of analytical pure grade and were purchased from Nanjing Chemical Reagent Co., Ltd. (Nanjing, China). Leaf quality and stem diameter determined the Chinese cabbage grade.

The experimental soil was an air-dried commercial seedling medium purchased from Shaanxi Yangling Yufeng Seed Industry Co., Ltd. (Xianyang, China). The cabbage seeds used were commercial Suzhou Qing seeds (Xingyun Vegetable Seed Breeding Center, Qing County, China). The seeds were germinated in a seedling box at a temperature of 20/25 °C (day/night), with humidity of 50% (ZLC-100D; Shuolian Equipment Co., Ltd., Hangzhou, China). Four weeks later, when the seedlings had three true leaves, uniform seedlings were selected for subsequent experiments. Every three seedlings were placed in a plastic pot (10 × 7 × 8.5 cm) with a drainage hole of 1 cm diameter at the bottom. A 20-mesh insect-proof net of 5 × 5 cm was placed on the bottom of the pot to prevent soil leakage.

### 2.1.1. Experiment I: Screening Experiment

In experiment I, we used a response surface model (Design Expert 9; Stat-Ease, Minneapolis, MN, USA) to determine the optimum N-P-K solution for Chinese cabbage growth (Table S1). In the model, the variates were as follows: unite 0 = 0 g/L, unite 1 = 0.5 g/L, unite 2 = 1 g/L, and unite 3 = 1.5 g/L. There were five replicates of each treatment, and the fertilizer treatment was applied to the seedlings every 7 days, with a total of four fertilizer applications and one control one. During the experiment, plant heights and stem diameters were recorded once a week. After four weeks, the plants were harvested. Nine seedlings were randomly selected from each treatment. The above- and belowground parts were then separated, and the stem diameters, root lengths, and above- and belowground fresh weights were measured. The root length was measured from the base of the stem to the tip of the main root. The plants were then transferred to a 105 °C oven for 30 min and then 80 °C for 24 h. The dry weights of the plants were then measured.

### 2.1.2. Experiment II

Based on the results of experiment I, the fertilizer treatments were compared to determine the best ratio of N-P-K solution with or without BNPs. In the protocol design (Table S2), the variates were as follows: unite 0 = 0 g/L, unite 1 = 0.5 g/L, unite 2 = 1 g/L, and unite 3 = 1.5 g/L. There were five replicates of each treatment. In experiment II, the fertilizer treatments (Table S2) were added to the plants every 2 weeks. Four weeks later, the Chinese cabbage plants were harvested, and the plant heights, root lengths,

above- and belowground fresh weights, stem diameters, and dry weights were recorded for further analysis.

### 2.2. Content of Elements

The dried above- and belowground tissues were ground and passed through a 1 mm sieve for elemental analysis. Each plant tissue sample (0.5 g) was digested in 50 mL polypropylene digestion tubes with 5 mL of nitric acid for 45 min on a heat block at 115 °C. The K, P, calcium (Ca), magnesium (Mg), sodium (Na), and sulfur (S) contents of the plant tissues were determined by inductively coupled plasma emission spectroscopy (Agilent 710 Series; Santa Clara, CA, USA). The elemental content was expressed in mg kg$^{-1}$ (dry weight) plant tissue.

### 2.3. Statistical Analysis

Statistical analysis of the data was performed using SPSS 21.0 statistical software (IBM Crop., Armonk, NY, USA). The data with the replicates was performed for analysis with variance. The figures were constructed using PAST 4.03 (University of Oslo, Norway). Means were compared using the least significant difference test and Duncan's new multiple range test. In this study, statistical significance was set at the level of $p < 0.05$.

## 3. Results

### 3.1. The Properties of BNPs

TEM characterization of the prepared BNPs revealed that they were of uniform size (approximately 85 nm) (Figure 1c,d). In an XRD analysis, the BNPs showed higher peaks from 18° to 30°, pointing to higher carbon contents in the BNPs. Peaks observed from 15° to 36° revealed stacking of monoatomic carbon layers as crystallites in the BNPs (Figure S1). Furthermore, the XPS result presents that the BNPs are rich in oxygen-containing groups on the surface (Figure S2). Based on the surface-area analysis, it is suggested that the BNPs are rich in pores on the surface, and it can perform the absorption model with unrestricted monolayer-multiplayer absorption (Table S1).

### 3.2. Experiment I: The Optimum N-P-K Solution for Chinese Cabbage Growth

After 4 weeks of growth and four fertilizer applications, based on the treatments of response surface model, Chinese cabbage growth was best in the T4, T13, and T15 treatments with N-P-K ratios of 2-0-2, 1-1-2, and 2-1-1, respectively. Chinese cabbage growth was the worst in the T7 and T11 treatments with higher N-P-K ratios of 2-3-2 and 3-2-2, respectively, (Figure 1a).

In the control (T1), the plant size and leaf area of Chinese cabbage were the smallest in all of the treatments (Figure 1a). Furthermore, the Chinese cabbage above-ground part fresh biomass and dry biomass was the lowest, at 4.90 and 0.51 in the control compared with other treatments, respectively (Figure 1d,f). However, the Chinese cabbage under-ground part presented the highest fresh biomass and dry biomass, which reached 1.30 and 0.22 in the control and other treatments, respectively (Figure 1e,g).

Plant height and root length reached 12.05 and 16.23 cm, respectively, in the T4 treatment. The maximum increase of 0.05 cm in stem dimeter was found in the T5 treatment. The T13 and T15 treatments also showed trends toward increases in plant heights, root lengths, and stem dimeters (11.56, 14.18, and 3.04 cm, respectively) as compared with the control after 4 weeks. The plants in the T7, T8, T9, T10, and T11 treatments showed a decreasing trend in plant heights, root lengths, and stem dimeters. In addition, root lengths in the T3, T6, and T7 treatments exhibited a decreasing trend (Figure 2a–c).

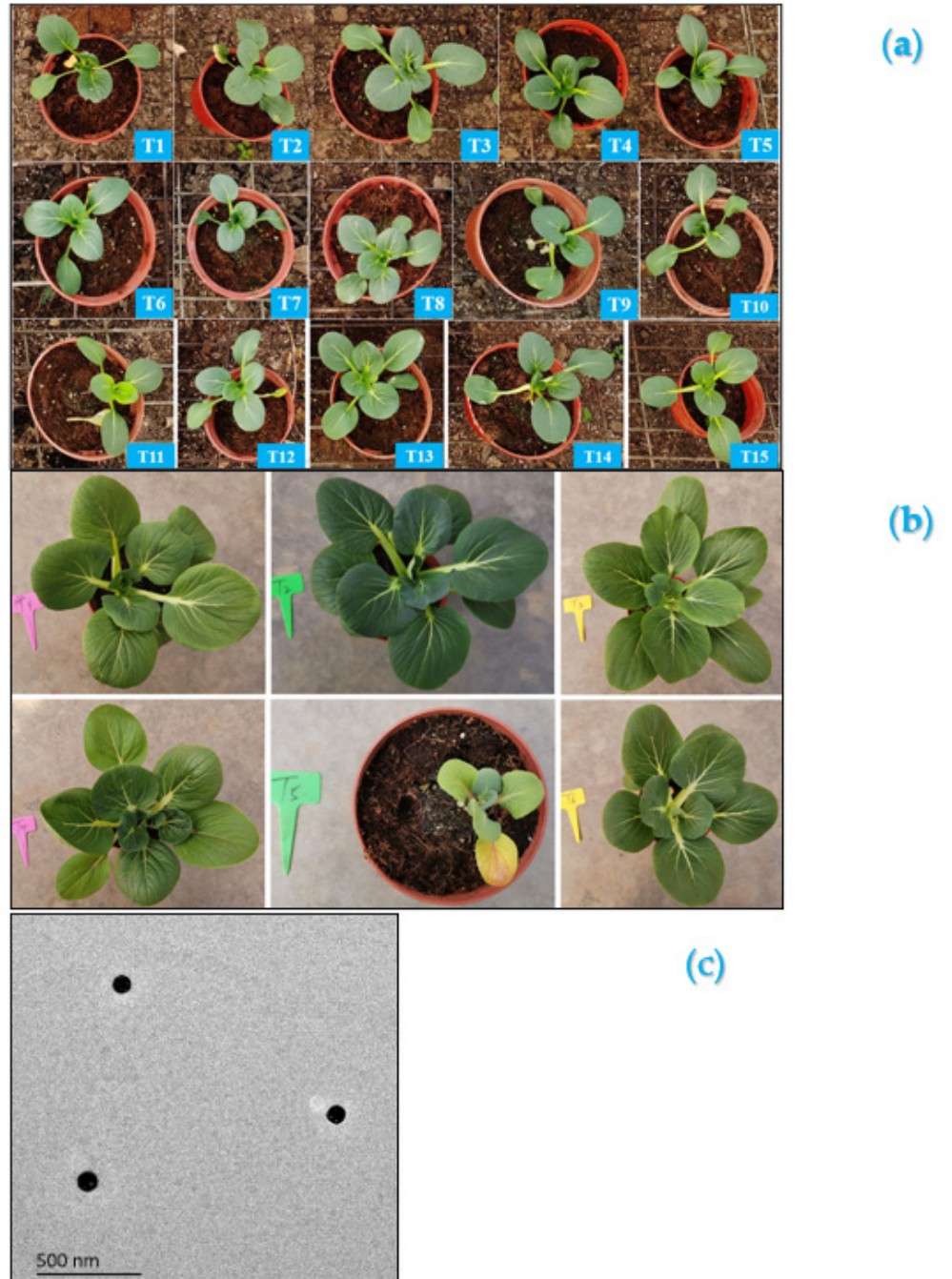

**Figure 1.** The phenotype of Chinese cabbage in the screening experiment after four weeks (**a**); and the phenotype of Chinese cabbage in the comparison experiment after four weeks (**b**). The BNPs TEM image (**c**). (Note, in Figure a, T1: N-P-K=0-0-0, T2: N-P-K=0-2-2, T3: N-P-K=1-2-2, T4: N-P-K=2-0-2, T5: N-P-K=2-1-2, T6: N-P-K=2-2-2, T7: N-P-K=2-3-2, T8: N-P-K=2-2-3, T9: N-P-K=2-2-0, T10: N-P-K=2-2-1, T11: N-P-K=3-2-2, T12: N-P-K=1-3-2, T13: N-P-K=1-1-2, T14: N-P-K=1-2-1, T15: N-P-K=2-1-1; in Figure 2, T1: BNPs-N-K=1:2:2, T2: BNPs-N-K=2:2:2, T3: BNPs -N-K=2:1:1, T4: BNPs-N-K=2:0.5:0.5, T5: BNPs-N-K=2:0:0, T6: BNPs-N-K=0:2:2; BNPs, biochar nanoparticles; 1 unit means 0.5 g/L nutrient).

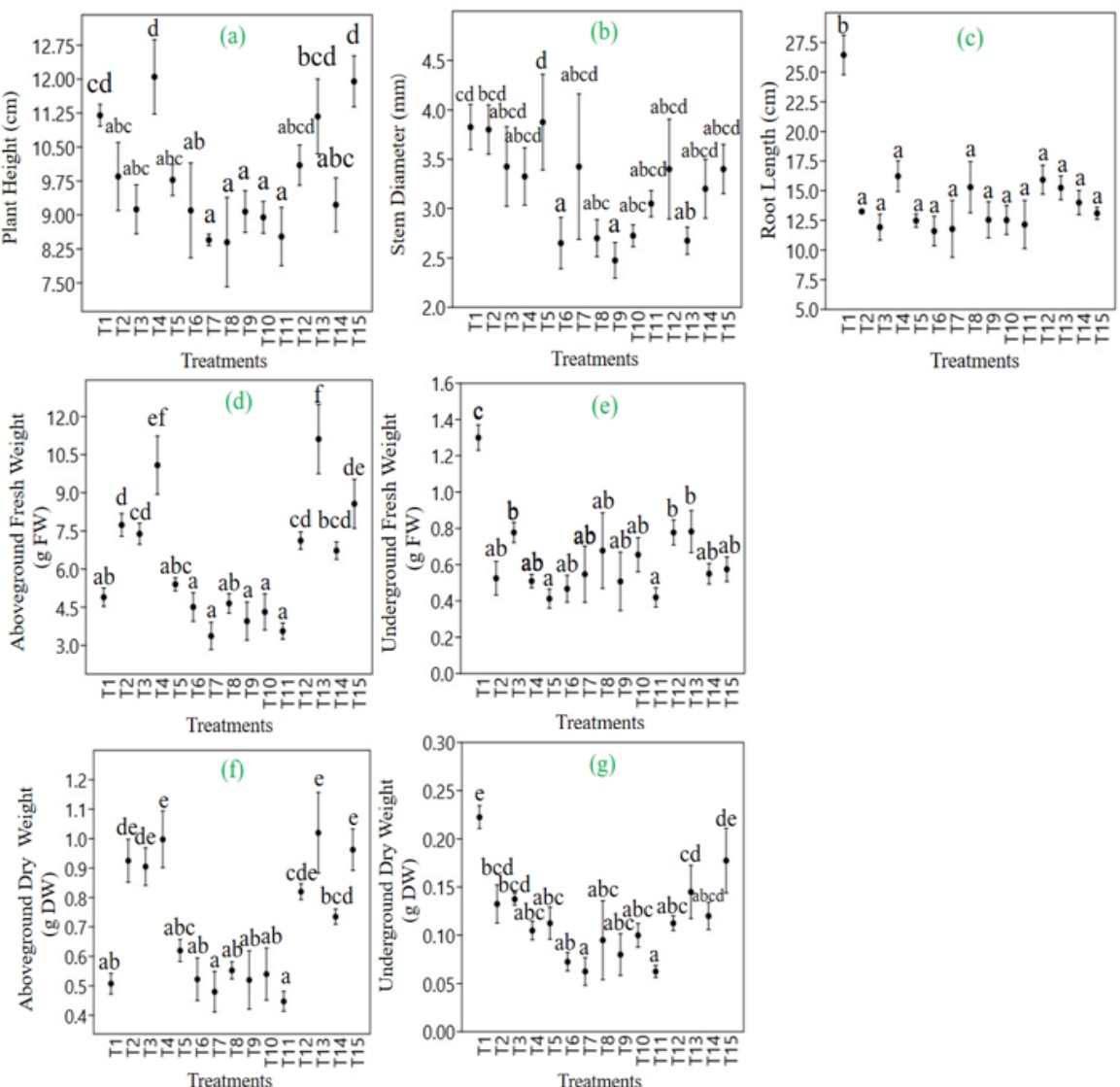

**Figure 2.** The ration effects of nitrogen, phosphorus, and potassium on plant height (**a**), stem diameter (**b**), root length (**c**), fresh weight (**d,e**) and dry weight (**f,g**) after four weeks treatments. (Note, T1: N-P-K=0-0-0, T2: N-P-K=0-2-2, T3: N-P-K=1-2-2, T4: N-P-K=2-0-2, T5: N-P-K=2-1-2, T6: N-P-K=2-2-2, T7: N-P-K=2-3-2, T8: N-P-K=2-2-3, T9: N-P-K=2-2-0, T10: N-P-K=2-2-1, T11: N-P-K=3-2-2, T12: N-P-K=1-3-2, T13: N-P-K=1-1-2, T14: N-P-K=1-2-1, T15: N-P-K=2-1-1; 1 unit means 0.5 g/L nutrient; Error bars indicate standard error of the mean; Different letters indicate the significant differences with $p < 0.05$.).

Among all the treatments, the fresh weights of the aboveground parts of Chinese cabbage were the best in the T4 and T13 treatments, with weights of 10.09 and 11.11 g, respectively. However, in terms of dry weights, the belowground parts of the control treatment were the highest (0.22 g). The average of fresh weight and dry weight in the T7, T9, T10, and T11 treatments was approximately 3.80 and 0.50 g in the Chinese cabbage, respectively, and the fresh and dry weights in these treatments were significantly lower than those in the other treatments ($p < 0.05$) (Figure 2c,e). Based on the plant phenotype, height, root length, and biomass, the optimum solution of N-P-K was 1.0-0-1.0 g/L (T4, 2-0-2).

### 3.3. Experiment II: The Application of BNPs with N-P-K for Chinese Cabbage Growth

After milling, corn BNPs with a diameter of 85 nm were obtained (Figure 1c,d). These BNPs were applied in experiment II. Based on the results of experiment I, 2 g/L of N, and

2 g/L of K were mixed with BNPs in experiment II to find out the best formula of BNPs and fertilizers. After 4 weeks treatments, Chinese cabbage showed the best phenotype in the T2 treatment (2 g/L of BNPs, 2 g/L of N, and 2 g/L of K), with a brighter green color and better growth than the other treatments. Plant growth in the experimental control (BNPs only) was poor (T5). In the T5 treatment, all the leaves had chlorosis. Plant size was smaller in the N-P-K-only treatment (T4 in Experiment I) when compared with that in the N-P-K plus BNP treatment (T6).

After the addition of the BNPs, Chinese cabbage height and stem diameter were 1.25 and 0.63 times greater, respectively, than the height and stem diameter of the control (T5). The treatment (T2) to which the BNPs and fertilizer were added resulted in an increase of 2.23 cm and 1.31 cm in plant height and stem diameter, respectively, as compared to that of the T5 (Figure 3a,b). However, there was no significant difference in the root length among these treatments ($p < 0.05$) (Figure 3c).

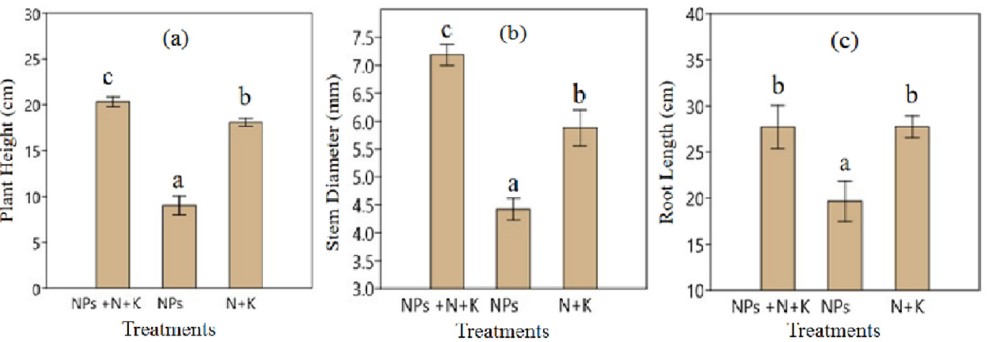

**Figure 3.** The ration effects of biochar nanoparticles, nitrogen, and potassium on Chinese cabbage of height (**a**), stem diameter (**b**) and root length (**c**) after four weeks treatments. (Note, NPs+N+K: BNPs-N-K=1:2:2 (T1), NPs: BNPs-N-K=2:0:0 (T5), N+K: BNPs-N-K=0:2:2 (T6); BNPs, biochar nanoparticles; 1 unit means 0.5 g/L nutrient; Error bars indicate standard error of the mean; Different letters indicate the significant differences with $p < 0.05$).

The fresh weight of the aboveground plant parts significantly increased after the addition of BNPs ($p < 0.05$). The aboveground weight reached 52.60 g after 4 weeks in the T2, with a significant increase in the dry weights of the above- and belowground parts (3.66 g and 0.52 g, respectively) ($p < 0.05$) as compared with that of the control treatment (Figure 4a,c,d). However, Chinese cabbage showed no significant increase, with or without the addition of BNPs (Figure 4b).

In the elemental analysis, there was no significant difference in the K content in the treatments with or without the addition of BNPs or in the treatments with or without the addition of fertilizers ($p < 0.05$). The average K content of above- and belowground parts was 28.45 g/kg and 46.93 g/kg, respectively, (Figure 5a,d). The P content of the above- and belowground parts of Chinese cabbage were significantly higher in the BNPs-only treatments than in T1 treatment ($p < 0.05$); however, the fertilized treatments were at the lower level, with contents of 2.13 and 1.81 g/kg, respectively (Figure 5b,e). The Ca content of Chinese cabbage was higher in the fertilizer treatments than in the T5 (Figure 5c,f). There was no significant difference in the Mg contents of Chinese cabbage leaves compared with the plants in T2 and the control ($p < 0.05$). The Mg content of underground parts was significantly lower in T6 (1.92 g/kg) than those treated in T5 (Figure 5g,j). The Na content of the aboveground parts was significantly higher in the BNP-only treatments than in the BNP plus fertilizer treatments. The Na content of the belowground parts was significantly higher in the BNPs plus fertilizer treatments ($p < 0.05$) (Figure 5h,k). The S contents were significantly lower only in the BNP treatments (T5), with contents of 4.44 and 3.56 g/kg, respectively (Figure 5i,l).

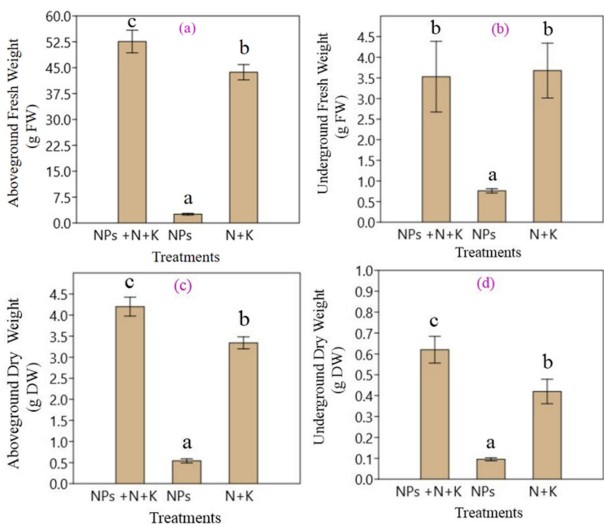

**Figure 4.** The ration effects of biochar nanoparticles, nitrogen, and potassium on Chinese cabbage of fresh (**a**,**b**) and dry (**c**,**d**) weight of plants after four weeks treatments. (Note, NPs+N+K: BNPs-N-K=1:2:2 (T1), NPs: BNPs-N-K=2:0:0 (T5), N+K: BNPs-N-K=0:2:2 (T6); BNPs, biochar nanoparticles; 1 unit means 0.5 g/L nutrient; Error bars indicate standard error of the mean; Different letters indicate the significant differences with $p < 0.05$.).

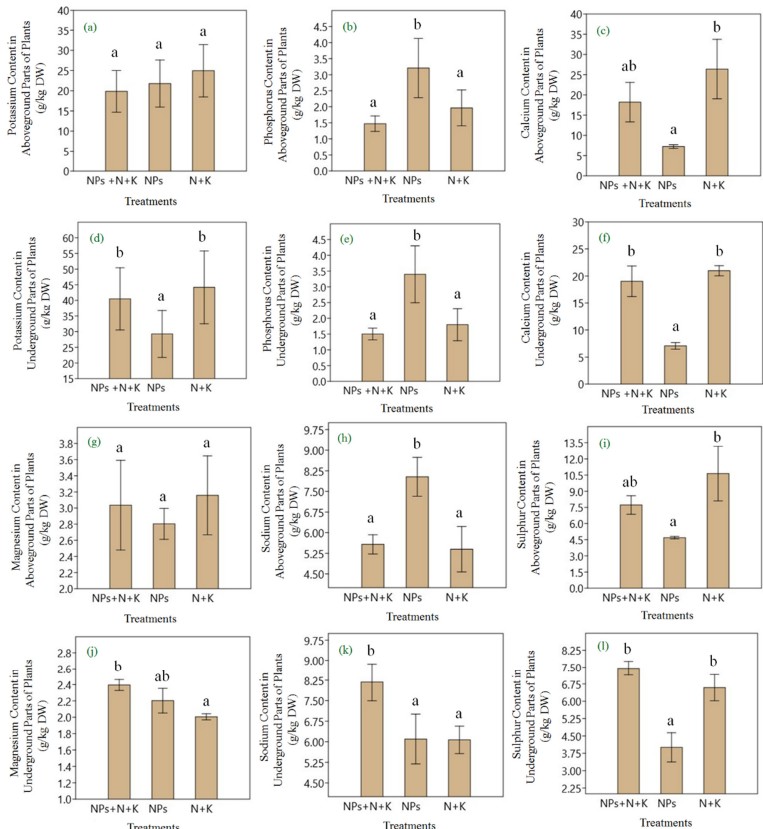

**Figure 5.** The ration effects of biochar nanoparticles, nitrogen, and potassium on Chinese cabbage of K (**a**,**d**), P (**b**,**e**), Ca (**c**,**f**), Mg (**g**,**j**), Na (**h**,**k**), and S (**i**,**l**) content of plants. (Note, NPs+N+K: BNPs-N-K=1:2:2 (T?), NPs: BNPs-N-K=2:0:0 (T5), N+K: BNPs-N-K=0:2:2 (T6); BNPs, biochar nanoparticles; 1 unit means 0.5 g/L nutrient; Error bars indicate standard error of the mean; Different letters indicate the significant differences with $p < 0.05$.)

## 4. Discussion

### 4.1. N-P-K Nutrient Management for Chinese Cabbage Growth

It is reported that N and P are important for plant growth [27]. In experiment I, we found that there is no positive correlation between plant growth and an increase in fertilizer applications. In this study, P was not a significant element for Chinese cabbage growth (Figures 1 and 2). Previous research reported that rhizosphere colonization by phosphate-solubilizing bacteria enhanced Chinese cabbage growth by 7.21%, without the addition of P fertilizer [28]. It was reported that current agricultural production systems require high quantities of P fertilizers, which have led to a build-up of legacy-P in soils [29]. The left P has become another P source for plant use; the rhizobacterium *Proteus vulgaris* JBLS202 stimulated Chinese cabbage growth, with an increase of 32.6% [30]. In a study that used green fluorescent protein to study YL6 colonization of Chinese cabbage roots, the biomass of the colonized plants increased 400% as compared with that of noncolonized plants [31]. It was reported that N additions did not significantly improve Chinese cabbage or maize growth [32], and we found that N has no significant relationship with the Chinese cabbage growth (Figure 1a). Our findings are in accordance with those of this previous study.

As reported previously, the application of phosphate-solubilizing bacteria can reduce the consumption of fertilizer and aid sustainable agricultural development. Exogenous additions of N and K are necessary for Chinese cabbage growth, but P is not. This is the intended reason that P is missing from experiment II.

### 4.2. The Effects of BNPs on Chinese Cabbage Growth

In experiment II, the BNPs significantly improved Chinese cabbage growth, especially the aboveground parts (heights, fresh and dry weights, and stem diameters) (Figures 1b and 2). Chinese cabbage was significantly smaller and lighter in the N-K-only fertilizer treatments as compared with the N-K with BNPs treatments. These results indicate that BNPs and N-K fertilizers applied in a two-time fertilizer application regime led to slow release of fertilizers and therefore increase the efficiency of the fertilizer treatment as compared with traditional applications of fertilizers. Compared to the fresh weights and heights of Chinese cabbage in T2, those in T5 and T6 were reduced 718.10 and 107.45%, respectively, (Figure 2). In terms of the efficiency of BNPs as a nanocarrier for fertilizers, BNPs improved the biomass harvest by 18.49% as compared to the treatments without BNPs. As compared with the other treatments, the N-K treatment combined with BNPs increased the biomass of the aboveground parts of Chinese cabbage. As reported previously, using nanocarriers for fertilizers can increase plant height because they have greater potential to provide plant nutrients continuously than conventional fertilizers [33]. Compared with the fertilizer-only treatments, the fertilizer plus BNP treatments significantly increased the dry matter biomass. The results suggest that BNPs are an efficient nanocarrier for fertilizers.

In addition, we found the BNPs on their own did not have positive effects on Chinese cabbage growth (Figure 1b), with Chinese cabbage treated with BNPs showing dwarfism and chlorosis. This result confirmed that BNPs do not provide nutrients for plant growth. Thus, we suggest that BNPs function only as a nanocarrier for fertilizers.

As we know, Mg contributes to plant growth and photosynthesis, and it is important for leaf development [34]. Higher Mg contents lead to higher fresh and dry weights of vegetables [35]. Na is a macronutrient required for plant growth. Na has been reported to increase WUE and stomatal diffusion and augment carbon dioxide uptake efficiency, thereby resulting in significant gains in nutritional status and positive plant physiological responses [36]. In the present study, in the treatments containing BNPs, more Mg and Na were transferred from the soil and the plant roots to the aboveground parts, leading to an increase in fresh biomass (Figure 5). In addition, Chinese cabbage quality increased in accordance with an increase of Mg and Na in the N-K-loaded BNPs treatments. As the fourth major plant nutrient, S, a constituent of three S-containing amino acids, participates in the formation of chlorophyll for photosynthesis [37]. In this study, the addition of BNPs reduced the uptake of S from the roots. However, the accumulation of S in the roots had no

effects on the aboveground parts of the plant. Therefore, we suggest that BNPs can increase Chinese cabbage absorption of Mg and Na while not limiting aspects of other elements. Additionally, due to the properties of biochar, P contents are around 60 to 85 ppm in the biochar [38], and this is the reason that Chinese cabbage can have more P in the above- and under-ground parts. In addition, the added biochar can help plants to absorb P [39], and it makes P more effective for plant absorption.

## 5. Conclusions

BNPs have been applied in agriculture. In a previous study, we applied BNPs to alleviate the allelopathic effects caused from *Imperata cylindrica* on rice growth [26], and it was the first time that we applied the BNPs working as an immunity promoter to induce the disease resistance in *Nicotiana benthamiana* [40]. Furthermore, extend research studied the application of BNPs in agriculture. Using nano-enabled fertilizers and nanocarriers, such as BNPs, fertilizers can be formulated or "tuned" to release nutrients in a controlled manner [41]; Figure 6 presents the potential mechanism of BNPs working as nanocarrier for Chinese cabbage. Using this "smart" technology enables a slower release of fertilizers, which can help plants absorb nutrients more efficiently than traditional fertilizers, improving nutrient usage at the same time. The present study demonstrates that using BNPs as a nanocarrier can improve plant nutrient absorption and fertilizer efficiency and improve the phenotype and quality of the plant in terms of biomass, height, and root length. In this study, the application of the BNP treatment to the roots increased the transfer of Mg and Na for plant growth while not affecting the transfer of other elements. Thus, BNPs can be considered a nanocarrier for the application of fertilizers in agriculture.

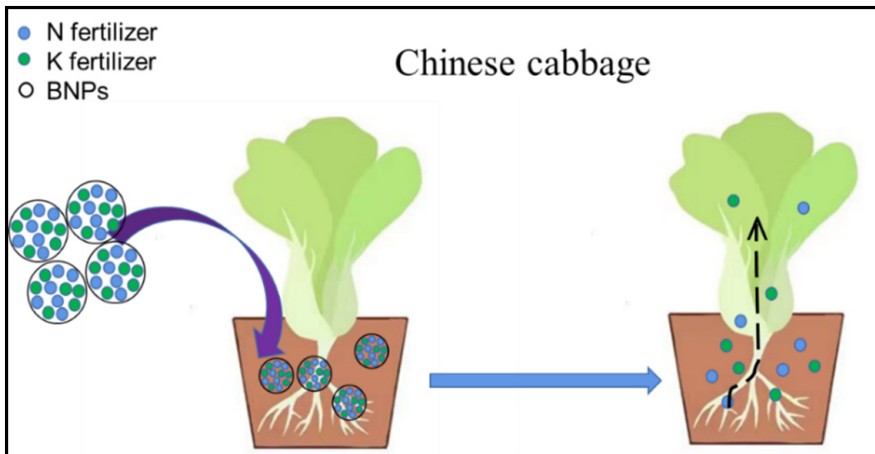

**Figure 6.** The potential mechanism of BNPs working as nanocarriers to regulate Chinese cabbage growth.

This study provides the use of BNPs as a nanocarrier that can better to manage the fertilizer release than traditional fertilizer treatments, thereby benefiting Chinese cabbage growth. Soil application of BNPs has benefits for Chinese cabbage in terms of plant growth, plant height, stem diameter, dry mass, and fresh biomass. Using only half the amount of fertilizers (2 g/L of BNPs and N-P-K) could achieve the same target in terms of Chinese cabbage growth as twice the amount of fertilizers applied using traditional methods. In this study, compared with the fertilizer treatments alone, Chinese cabbage height increased more than 18–25% and the fresh biomass increased 44% in the BNP plus fertilizer treatments. We conclude that BNPs provide a novel and efficient nutrient delivery method to improve plant growth, which is essential in agriculture to achieve more sustainable crop systems. This study demonstrates the potential of nanocarriers in agricultural fertilizer applications.

**Supplementary Materials:** The following supporting information can be downloaded at: https://www.mdpi.com/article/10.3390/coatings12121984/s1, The BNPs physical properties of The X-ray diffractometer (XRD), X-ray photoelectron spectroscopy (XPS), and the surface-area and the pore size were present in the Figures S1, S2 and Table S1, respectively. Response Surface Model information was presented in Table S2. The Treatments for Experiment of Comparison was presented in Table S3. Plant weight and Nutrient contents for Experiment of Comparison were shown in Tables S4 and S5.

**Author Contributions:** Conceptualization, R.Y. and Y.S.; methodology, Y.S.; formal analysis, R.Y.; investigation, J.S., Y.Z. and L.J.; data curation, R.Y., Z.W. and X.S.; writing—original draft preparation, R.Y.; writing—review and editing, Y.S.; visualization, R.Y. and J.S.; supervision, B.T. and Y.S.; project administration, B.T.; funding acquisition, R.Y. and Y.S. All authors have read and agreed to the published version of the manuscript.

**Funding:** This research was supported by the Primary Research & Development Plan of Jiangsu Province [BE2020673]; the Innovation Project of Jiangsu Academy of Agricultural Science and Technology [SCX(22)3985], and National Key R&D Program of China [2019YFD0900404-05].

**Institutional Review Board Statement:** Not applicable.

**Informed Consent Statement:** Not applicable.

**Data Availability Statement:** Not applicable.

**Conflicts of Interest:** The authors declare no competing financial interests.

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
