# Peer review of "The Role of Biochar Nanoparticles Performing as Nanocarriers for Fertilizers on the Growth Promotion of Chinese Cabbage (Brassica rapa (Pekinensis Group))"

_coatings, doi:10.3390/coatings12121984_

Round 1
Reviewer 1 Report
The authors have made great efforts to evaluate the effects of biochar nanoparticles on the growth of Chinese cabbage (Brassica rapa). The attained results are of interest for readers, agronomists to develop environmentally-friendly sustainable agriculture. However, the current form of manuscript has not been well organized and documented. Much information should be specifically provided. The authors can find some useful information, comments and suggestions to improve the quality of the manuscript (see detail in pdf file):
1. Abstract must be reworded
2. Introduction: line 54-55, Should add the equal value in USD for wide readers understanding; line 61-63: This sentence seems the contradict meaning, hence it should be reworded and added some examples of advantages of the fertilizers. The specific objectives must be mentioned at the end of this part
3. Materials and methods: This part must be judiciously narrated, and should not mention the attained results in this part (see in pdf file)
4. Fig 1, a, b, c, d did not marked in Fig, recheck
5. The different treatments such as T1, T2, T3....T11 must be described in Materials and Methods part.
6. in all sub-Figures, what dose it mean of the letter "a, b, c, d, e, f...etc?, should take a note. The specific value of error bars must be added (see comments in pdf file).
7. In Figs, and supplementary information, the statistical data analyses were not found, even though some software analyses were performed, check please
8. English editing should be made, there have been many spellings, typing and grammar errors found in the current form of the manuscript (see pdf file).
9. Discussion should be improved, is there any information available by using BNPs for other crops, vegetables?
10. Conclusion must be reworded, need to explain or make reference cited, only mentioned what is the most significant finding of this study
11. Reference part must be edited and rechecked (see in pdf file)

Author Response
A letter to the Editor of Coatings,
Journal: Coatings
Manuscript ID: Coatings-2033593.R1
Title: " The role of biochar nanoparticles performing as nanocarrier for fertilizers on the growth promotion of Chinese cabbage (Brassica rapa (Pekinensis Group))"
Author(s): Xiaoping Sun, Jiamin Shen, Yu Shen, Ran Tao, Mingchang Cao and Yang Xiao
Dear Editor,
Please find our revised manuscript Coatings-2033593.R1 as well as a series of point-by-point responses to the reviewer comments. We note that in this revised manuscript, we have fully addressed all comments by the three reviewers. We’d like to thank the reviewers for their constructive comments and thoughtful evaluation of our work. In addition to point-by-point responses below, we also provide clean and “track-changes” versions of the revised manuscript. The corresponding changes are made and highlighted in red in the main text. We feel that these changes have significantly improved the paper and now hope that it is acceptable for publication in the Coatings.
If you have any questions, please don’t hesitate to contact me.
Sincerely,
Yu Shen, Ph.D., Professor,
College of Biology and the Environment,
Nanjing Forestry University, Nanjing 210037, China;
E-mail: sheyttmax@hotmail.com/yushen@njfu.edu.cn.
Reviewer #1
General Comments
The authors have made great efforts to evaluate the effects of biochar nanoparticles on the growth of Chinese cabbage (Brassica rapa). The attained results are of interest for readers, agronomists to develop environmentally-friendly sustainable agriculture. However, the current form of manuscript has not been well organized and documented. Much information should be specifically provided. The authors can find some useful information, comments and suggestions to improve the quality of the manuscript (see detail in pdf file).
Answer: Thanks very much for your kind suggestions and guidance. We have revised our manuscript carefully one by one point. Please check it.
Question 1. Abstract must be reworded
Answer: Thanks for your careful check. We have revised our Abstract in the resubmitted manuscript.
Question 2. Introduction: line 54-55, Should add the equal value in USD for wide readers understanding; line 61-63: This sentence seems the contradict meaning, hence it should be reworded and added some examples of advantages of the fertilizers. The specific objectives must be mentioned at the end of this part
Answer: Thanks very much for your good suggestions. We have switched the currency unite from RMB to US dollar in the revised manuscript on line 55. And due to the common knowledge of the fertilizers, we added some briefly advantages of fertilizer use in agriculture, and pointed out the research target with red words in the revised manuscript.
Question 3. Materials and methods: This part must be judiciously narrated, and should not mention the attained results in this part (see in pdf file)
Answer: Thanks very much for your good suggestions. We revised 2.1 Biochar NPs preparation part, and put the detailed result in 3.1 The properties of BNPs.
Question 4. Fig 1, a, b, c, d did not marked in Fig, recheck
Answer: Thanks for your check. It came a mistake in the submission of the marked letters in the Figure 1, and we revised the letters with blue letters in the revised manuscript to make them clearly.
Question 5. The different treatments such as T1, T2, T3....T11 must be described in Materials and Methods part.
Answer: Thanks for your good suggestion. We have added the treatment information of the fertilizer methods of Experiment I in the revised manuscript.
Question 6. in all sub-Figures, what dose it mean of the letter "a, b, c, d, e, f...etc?, should take a note. The specific value of error bars must be added (see comments in pdf file).
Answer: Thanks for your careful check. We added the description of the letters in not of the Figure citation: “Error bars indicate standard error of the mean; Different letters indicate the significant differences with P < 0.05.”
Question 7. In Figs, and supplementary information, the statistical data analyses were not found, even though some software analyses were performed, check please
Answer: Thanks for your good suggestion to improve our manuscript quality. We added the data analysis in Table S4 Plant weight for Experiment of Comparison and Table S5 Nutrient contents for Experiment of Comparison
Question 8. English editing should be made, there have been many spellings, typing and grammar errors found in the current form of the manuscript (see pdf file).
Answer: Thanks very much for your kind guidance. We have checked the spelling, typing and grammar in the revised manuscript. Please check it.
Question 9. Discussion should be improved, is there any information available by using BNPs for other crops, vegetables?
Answer: Thanks very much for your good comment. BNPs has been used for the environmental remediation, but it is little known that how it can be applied in the agriculture directly. In previous study, we applied the BNPs to alleviate the allelopathic effects caused from Imperata cylindrica on the rice growth (Shen et al., 2020); and it was the first that we applied the BNPs working as an immunity promoter to induce the disease resistance in Nicotiana benthamiana (Shen et al., 2022). And we have added the application of BNPs in the revised manuscript.
Reference
Shen Y, Tang H, Wu W, et al. Role of nano-biochar in attenuating the allelopathic effect from Imperata cylindrica on rice seedlings[J]. Environmental Science: Nano, 2020, 7(1): 116-126.
Kong M, Liang J, White J C, et al. Biochar nanoparticle-induced plant immunity and its application with the elicitor methoxyindole in Nicotiana benthamiana[J]. Environmental Science: Nano, 2022, 9(9): 3514-3524.
Question 10. Conclusion must be reworded, need to explain or make reference cited, only mentioned what is the most significant finding of this study
Answer: Thanks very much for your good suggestion to improve our manuscript quality. We added some more information about the application of NPs as fertilizer management in agriculture in the revised manuscript.
Question 11. Reference part must be edited and rechecked (see in pdf file)
Answer: Thanks very much. We have revised the reference types in the revised manuscript.

Reviewer 2 Report
Coatings-2033593 Comment
The MS are interested in the nano-biochar that supports Chinese cabbage growth without the use of P fertilizer.
Major remark
From p5 line 247- : Our results agree with previous findings that phosphate-solubilizing bacteria can aid Chinese cabbage growth by increasing the P content …
1. So, the MS should be specific about the soil/supporting materials that is involved with any P-microbial, etc.
2. How does biochar provide phosphorus without rhizosphere organism? The Information should be included in the discussion. (See ex. https://www.nature.com/articles/s41598-019-45693-z)
3. Explain reason why the NPs treatment had a higher P concentration than the others (N+K and NPs+N+K) in Figure 5.
From p6 line 249- : Exogenous additions of N and K are necessary for Chinese cabbage growth, but P is not. This is the potential reason that P is 250 missing from experiment II.
4. The discussion of "growth" appears hazy and exaggerated. Other growth parameters were missing, such as leaf thickness, leaf size, and leaf greenness. As a result, please address it directly in each growth parameter in this study.
5. In general, the availability of phosphorus is important for sustainable use in https://doi.org/10.1007/s42729-022-00980-z and “Biochar is a potential material for making slow-releasing phosphorus (P) fertilizers for the sake of increasing soil P use efficiency and mitigating P losses” from https://doi.org/10.1016/j.chemosphere.2019.125471. N/P supported Chinese cabbage growth in https://www.cropandweed.com/archives/2009/vol5issue2/19.pdf . Including P enrichment by NPs in Fig” 5
These details, and so on, can be used in the MS for discussion editing for the function of biochar and P for the cabbage.
Minor Remark
1. Tables 1 and 2 have the correct wording in the main text. The main text could not be found.
2. Figure 1 (and so on): Reduced confusion by using different alphabet abbreviations in each treatment [Experiment I (screening experiment) and Experiment II (comparison experiment)], such as T1...,...C1..., and so on.
3. Figure 1 (and so on): What is the difference between c and d in the biochar nanoparticle TEM image? In the figure legend, include a brief indication.
4. Figures 4, 5 and a story involving: The figures had already written the NPs+N+K, NPs, N+K. Correct the abbreviation note and detail in the MS (such as line 225-227 etc.).
5. Isn't that phosphorus in Fig. 5 e? Please check the MS for details as well.
6. From Line 300-301: "This study shows that using BNPs as a nanocarrier results in slower fertilizer release than traditional fertilizer treatments, which benefits Chinese cabbage growth..." Please be cautious about the sentence, as there is no direct result that shows a slow release of information.
………………………………………………………………………………………

Author Response
A letter to the Editor of Coatings,
Journal: Coatings
Manuscript ID: Coatings-2033593.R1
Title: " The role of biochar nanoparticles performing as nanocarrier for fertilizers on the growth promotion of Chinese cabbage (Brassica rapa (Pekinensis Group))"
Author(s): Xiaoping Sun, Jiamin Shen, Yu Shen, Ran Tao, Mingchang Cao and Yang Xiao
Dear Editor,
Please find our revised manuscript Coatings-2033593.R1 as well as a series of point-by-point responses to the reviewer comments. We note that in this revised manuscript, we have fully addressed all comments by the three reviewers. We’d like to thank the reviewers for their constructive comments and thoughtful evaluation of our work. In addition to point-by-point responses below, we also provide clean and “track-changes” versions of the revised manuscript. The corresponding changes are made and highlighted in red in the main text. We feel that these changes have significantly improved the paper and now hope that it is acceptable for publication in the Coatings.
If you have any questions, please don’t hesitate to contact me.
Sincerely,
Yu Shen, Ph.D., Professor,
College of Biology and the Environment,
Nanjing Forestry University, Nanjing 210037, China;
E-mail: sheyttmax@hotmail.com/yushen@njfu.edu.cn.
Reviewer #2
General Comments
The MS are interested in the nano-biochar that supports Chinese cabbage growth without the use of P fertilizer.
From p5 line 247- : Our results agree with previous findings that phosphate-solubilizing bacteria can aid Chinese cabbage growth by increasing the P content …
Answer: Thanks for your interests of our research. And it is another research that Chinese cabbage can absorb P from the soil directly and no need for the exogenous P addition from the fertilizer.
Question 1: So, the MS should be specific about the soil/supporting materials that is involved with any P-microbial, etc.
Answer: Thanks very much. We added the information about the soil and biochar supporting P absorption; as well as the description of rhizospheric microorganism assisting plant absorb P.
Question 2: How does biochar provide phosphorus without rhizosphere organism? The Information should be included in the discussion. (See ex. https://www.nature.com/articles/s41598-019-45693-z)
Answer: Thanks very much for your good suggestion to improve manuscript quality. It was reported that current agricultural production systems require high quantities of P fertilizers, which have led to a build-up of legacy-P in soils (El Attar et al., 2022). And the left P has become another P source for plant use. And the biochar added can help plants to absorb P (Glaser and Lehr, 2019), and it makes P be more effective for plant absorption. And we added the reason in the revised manuscript.
Reference
El Attar I, Hnini M, Taha K, et al. Phosphorus Availability and its Sustainable Use[J]. Journal of Soil Science and Plant Nutrition, 2022: 1-13.
Glaser B, Lehr V I. Biochar effects on phosphorus availability in agricultural soils: A meta-analysis[J]. Scientific reports, 2019, 9(1): 1-9.
Question 3: Explain reason why the NPs treatment had a higher P concentration than the others (N+K and NPs+N+K) in Figure 5.
Answer: Thanks very much for your good suggestion to improve our manuscript quality. You’re pretty right that the biochar can offer the P for the plant absorb when the soil application. Due to the properties of biochar, P contents are around 60 to 85 ppm in the biochar (Li et al., 2020); and this is one reason that the Chinese cabbage can have more P in the above- and under-ground parts. At the same time, it was reported that current agricultural production systems require high quantities of P fertilizers, which have led to a build-up of legacy-P in soils (El Attar et al., 2022). And the left P has become another P source for plant use. This is another reason for the P contents increase in the only BNPs treated Chinese cabbage.
Reference
El Attar I, Hnini M, Taha K, et al. Phosphorus Availability and its Sustainable Use[J]. Journal of Soil Science and Plant Nutrition, 2022: 1-13.
Li H, Li Y, Xu Y, et al. Biochar phosphorus fertilizer effects on soil phosphorus availability[J]. Chemosphere, 2020, 244: 125471.
Prasad P H, Bhunia P, Naik A, et al. Response of nitrogen and phosphorus levels on the growth and yield of chinese cabbage (Brassica campestris L. var. pekinensis) in the gangetic plains of West Bengal[J]. Journal of Crop and Weed, 2009, 5(2): 75-77.
Question 4. From p6 line 249- : Exogenous additions of N and K are necessary for Chinese cabbage growth, but P is not. This is the potential reason that P is missing from experiment II.
Answer: Thanks for your good comment. This is the reason that we conduct two experiments, the first one to screen the best fertilizer formula, and the second one is to improve the fertilizer efficiency.
Question 5. The discussion of "growth" appears hazy and exaggerated. Other growth parameters were missing, such as leaf thickness, leaf size, and leaf greenness. As a result, please address it directly in each growth parameter in this study.
Answer: Thanks very much for your kind suggestion. We applied the direct evidence to show the results of the Chinese cabbage growth; and the leaf size is one of the obvious factors in the plants. And we only use the size comparation directly, it is enough to know the leaf size differences between the Chinese cabbage in all treatments.
Question 6. In general, the availability of phosphorus is important for sustainable use in https://doi.org/10.1007/s42729-022-00980-z and “Biochar is a potential material for making slow-releasing phosphorus (P) fertilizers for the sake of increasing soil P use efficiency and mitigating P losses” from https://doi.org/10.1016/j.chemosphere.2019.125471. N/P supported Chinese cabbage growth in https://www.cropandweed.com/archives/2009/vol5issue2/19.pdf . Including P enrichment by NPs in Fig” 5.
These details, and so on, can be used in the MS for discussion editing for the function of biochar and P for the cabbage.
Answer: Thanks very much for your good suggestion to improve our manuscript quality. You’re pretty right that the biochar can offer the P for the plant absorb when the soil application. Due to the properties of biochar, P contents are around 60 to 85 ppm in the biochar (Li et al., 2020); and this is one reason that the Chinese cabbage can have more P in the above- and under-ground parts. At the same time, it was reported that current agricultural production systems require high quantities of P fertilizers, which have led to a build-up of legacy-P in soils (El Attar et al., 2022). And the left P has become another P source for plant use. This is another reason for the P contents increase in the only BNPs treated Chinese cabbage.
We have revised the discussion part. Please check it.
Reference
El Attar I, Hnini M, Taha K, et al. Phosphorus Availability and its Sustainable Use[J]. Journal of Soil Science and Plant Nutrition, 2022: 1-13.
Li H, Li Y, Xu Y, et al. Biochar phosphorus fertilizer effects on soil phosphorus availability[J]. Chemosphere, 2020, 244: 125471.
Prasad P H, Bhunia P, Naik A, et al. Response of nitrogen and phosphorus levels on the growth and yield of chinese cabbage (Brassica campestris L. var. pekinensis) in the gangetic plains of West Bengal[J]. Journal of Crop and Weed, 2009, 5(2): 75-77.
Minor Remark
Question 7. Tables 1 and 2 have the correct wording in the main text. The main text could not be found.
Answer: Thanks very much for your careful check to improve our manuscript quality. We revised the Table information in the manuscript.
Question 8. Figure 1 (and so on): Reduced confusion by using different alphabet abbreviations in each treatment [Experiment I (screening experiment) and Experiment II (comparison experiment)], such as T1...,...C1..., and so on.
Answer: Thanks very much for your good suggestion. We have revised the whole figures and tables design.
Question 9. Figure 1 (and so on): What is the difference between c and d in the biochar nanoparticle TEM image? In the figure legend, include a brief indication.
Answer: Thanks for your good comment. There are no differences of the biochar NPs between c and d in Figure 1. We only wanted to show some more images of the NPs. We deleted one image in the revised manuscript.
Question 10. Figures 4, 5 and a story involving: The figures had already written the NPs+N+K, NPs, N+K. Correct the abbreviation note and detail in the MS (such as line 225-227 etc.).
Answer: Thanks very much for your good suggestion. We update the writing of the treatments’ information in the revised manuscript.
Question 11. Isn't that phosphorus in Fig. 5 e? Please check the MS for details as well.
Answer: Thanks very much for your check to improve our manuscript quality. We checked and revised the elements information, and redesign the figures.
Question 12. From Line 300-301: "This study shows that using BNPs as a nanocarrier results in slower fertilizer release than traditional fertilizer treatments, which benefits Chinese cabbage growth..." Please be cautious about the sentence, as there is no direct result that shows a slow release of information. The manuscript explores the feasibility of using biochar nanoparticles as nanocarriers for fertilizers regarding the growth promotion of Chinese cabbage. The study has been conducted systematically. However, the following concerns should be addressed before further consideration for publication.
Answer: Thanks very much for your good comment. We revised our conclusion sentence that This study provides the use of BNPs as a nanocarrier can better to manage the fertilizer release than traditional fertilizer treatments, thereby benefiting Chinese cabbage growth.

Reviewer 3 Report
The manuscript explores the feasibility of using biochar nanoparticles as nanocarriers for fertilizers regarding the growth promotion of Chinese cabbage. The study has been conducted systematically. However, the following concerns should be addressed before further consideration for publication.
Literature survey should be improved, the studies regarding use of biochar nanoparticles as carriers for fertilizers should be discussed in detail and research gaps should be clearly defined. More recent references regarding use of biochar for agricultural and environmental applications should be included to illustrate the potential of the material. The following references may be useful: https://doi.org/10.1016/j.iswcr.2021.09.006 https://doi.org/10.1016/j.cherd.2022.07.043 The rationale of selecting a ratio of 15:1 for balls to powder should be justified. Control experiments should be more clearly described. Fig 1 c and d: what is the difference between the TEM images in two panels? Interpretation needs to be clarified In section 4.1, it is mentioned that N additions did not significantly improve Chinese cabbage growth. But toward the end of paragraph, a statement is made that exogenous additions of N and K are necessary for Chinese cabbage growth. Please check and clarify. English should be thoroughly checked.
Author Response
A letter to the Editor of Coatings,
Journal: Coatings
Manuscript ID: Coatings-2033593.R1
Title: " The role of biochar nanoparticles performing as nanocarrier for fertilizers on the growth promotion of Chinese cabbage (Brassica rapa (Pekinensis Group))"
Author(s): Xiaoping Sun, Jiamin Shen, Yu Shen, Ran Tao, Mingchang Cao and Yang Xiao
Dear Editor,
Please find our revised manuscript Coatings-2033593.R1 as well as a series of point-by-point responses to the reviewer comments. We note that in this revised manuscript, we have fully addressed all comments by the three reviewers. We’d like to thank the reviewers for their constructive comments and thoughtful evaluation of our work. In addition to point-by-point responses below, we also provide clean and “track-changes” versions of the revised manuscript. The corresponding changes are made and highlighted in red in the main text. We feel that these changes have significantly improved the paper and now hope that it is acceptable for publication in the Coatings.
If you have any questions, please don’t hesitate to contact me.
Sincerely,
Yu Shen, Ph.D., Professor,
College of Biology and the Environment,
Nanjing Forestry University, Nanjing 210037, China;
E-mail: sheyttmax@hotmail.com/yushen@njfu.edu.cn.
Reviewer #3
Question 1. Literature survey should be improved, the studies regarding use of biochar nanoparticles as carriers for fertilizers should be discussed in detail and research gaps should be clearly defined.
More recent references regarding use of biochar for agricultural and environmental applications should be included to illustrate the potential of the material. The following references may be useful: https://doi.org/10.1016/j.iswcr.2021.09.006 https://doi.org/10.1016/j.cherd.2022.07.043
Answer: Thanks very much for your encouragement. Please find our revised manuscript, as well as a series of point-by-point responses to the reviewer comments. We note that in this revised manuscript, we have fully addressed all comments. We’d like to thank the reviewers for their constructive comments and thoughtful evaluation of our work.
We read the “Wijitkosum S. Biochar derived from agricultural wastes and wood residues for sustainable agricultural and environmental applications[J]. International Soil and Water Conservation Research, 2022, 10(2): 335-341.” and “Jacob M M, Ponnuchamy M, Kapoor A, et al. Adsorptive decontamination of organophosphate pesticide chlorpyrifos from aqueous systems using bagasse-derived biochar alginate beads: Thermodynamic, equilibrium, and kinetic studies[J]. Chemical Engineering Research and Design, 2022, 186: 241-251.” And we add some more new references in our articles.
Question 2. The rationale of selecting a ratio of 15:1 for balls to powder should be justified.
Answer: Thanks for your good comment. We did a large experiment to get the ration of ZrO2 balls to powder was 15:1. With this ration, we can get the uniformed size of biochar NPs around 100 - 150 nm. And the size of biochar NPs is stable and uniform; and we have used the NPs for disease resistance in Nicotiana benthamiana (Kong et al., 2022) and rice protection (Shen et al., 2020).
Reference
Kong M, Liang J, White J C, et al. Biochar nanoparticle-induced plant immunity and its application with the elicitor methoxyindole in Nicotiana benthamiana[J]. Environmental Science: Nano, 2022, 9(9): 3514-3524.
Shen Y, Tang H, Wu W, et al. Role of nano-biochar in attenuating the allelopathic effect from Imperata cylindrica on rice seedlings[J]. Environmental Science: Nano, 2020, 7(1): 116-126.
Question 3. Control experiments should be more clearly described.
Answer: Thanks very much for your suggestion. We added more description of the plants under control (no treated) in the revised manuscript.
Question 4. Fig 1 c and d: what is the difference between the TEM images in two panels? Interpretation needs to be clarified
Answer: Thanks for your good comment. There are no differences of the biochar NPs between c and d in Figure 1. We only wanted to show some more images of the NPs. We deleted one image in the revised manuscript.
Question 5. In section 4.1, it is mentioned that N additions did not significantly improve Chinese cabbage growth. But toward the end of paragraph, a statement is made that exogenous additions of N and K are necessary for Chinese cabbage growth. Please check and clarify.
Answer: Thanks for your good comment. It was a study from Zhao et al. (2022) that N additions did not significantly improve Chinese cabbage growth. In our study, we found that the N addition could improve the Chinese cabbage growth based on our research, but the improvement was not significant. Based on the result, we think N is still necessary for the Chinese cabbage growth. There are two different studies.
Reference
Zhao H, Xie T, Xiao H, et al. Biochar-Based Fertilizer Improved Crop Yields and N Utilization Efficiency in a Maize–Chinese Cabbage Rotation System[J]. Agriculture, 2022, 12(7): 1030.
Question 6. English should be thoroughly checked.
Answer: Thanks very much for your good suggestion. We went through all the words and English writing in the revised manuscript.

Reviewer 4 Report
The manuscript covers very important aspects – improvement of efficiency of fertilizer use. I find the results very relevant and worthy of publication. However, the manuscript needs major revision.
Table S1 and Table 2 – there is a lack of that information.
The manuscript could be also improved by one Figure which could show obtained results in an easier way.
Introduction
Line 41 -43 – something is missing in that statement…
Line 80 – “platform” – replace word..
There are no aims or hypotheses. The reader should get information about what you did and why, what you expected, and what was your hypothesis. Please provide them…
Material and methods –
please provide one Figure which presents the experimental design, it is hard to follow your experiment just on the provided description. There are many experimental treatments, therefore it would be ideal to present both Exp. On one scheme, you did Exp. 1 and on the base of the obtained data you did Exp. 2. Please show the relationship between those two experiments.
Line 124-128 – should be placed after information from Line 114.
There is no Table S1 (Line 112) , and Supplementary data (Line 111).
Exp 1. And Exp. 2 - Information about units is not clear. Please rewrite them.
Table 1 – is missing (Line 131)
Table 2 – is missing (Line 145, 147)
Line 151 - Elemental analysis – maybe it is better – Content of elements
Did you analyze the content of nitrogen, why not?
3.2. Results
Please provide information about the level of increase or decrease of analyzed parameters compared to non-fertilized plants (T1). It would be good to indicate that increase in fertilizer use, not always results in higher production of plants’ biomass.
Please indicate the amount of N, P, and K (g/L) which resulted in the best plant growth.
Similarly, please provide information in Exp. 2 according to observed changes between treatment groups. Please, indicate which treatment group is your control group and how the treatment used impacted on obtained results.
Moreover, please use the same names for treatment groups, thus Fig 4 and 5 are not the same as used in the description of Exp. 2, placed in the manuscript part: material and methods. Please, check carefully all the figures description…
There is no clear information, that in exp 2 you did not add P. Please add it.
Did you check the content of elements in the soil after the experiment ends, so you could point out that BNPs significantly impact the outflow of elements?
Discussion
Line – 248 – you did not analyze the effect of phosphate-solubilizing bacteria, so please remove these statements.
Line 250 – “potential reason” – potential, not intended?
Author Response
A letter to the Editor of Coatings,
Journal: Coatings
Manuscript ID: Coatings-2033593.R1
Title: " The role of biochar nanoparticles performing as nanocarrier for fertilizers on the growth promotion of Chinese cabbage (Brassica rapa (Pekinensis Group))"
Author(s): Ruiping Yang, Jiamin Shen, Yuhan Zhang, Lin Jiang, Xiaoping Sun, Zhengyang Wang, Boping Tang, Yu Shen.
Dear Editor,
Please find our revised manuscript Coatings-2033593.R1 as well as a series of point-by-point responses to the reviewer comments. We note that in this revised manuscript, we have fully addressed all comments by the three reviewers. We’d like to thank the reviewers for their constructive comments and thoughtful evaluation of our work. In addition to point-by-point responses below, we also provide clean and “track-changes” versions of the revised manuscript. The corresponding changes are made and highlighted in red in the main text. We feel that these changes have significantly improved the paper and now hope that it is acceptable for publication in the Coatings.
If you have any questions, please don’t hesitate to contact me.
Sincerely,
Yu Shen, Ph.D., Professor,
College of Biology and the Environment,
Nanjing Forestry University, Nanjing 210037, China;
E-mail: sheyttmax@hotmail.com/yushen@njfu.edu.cn.
Reviewer #4
General comment:
The manuscript covers very important aspects – improvement of efficiency of fertilizer use. I find the results very relevant and worthy of publication. However, the manuscript needs major revision.
Answer: Thanks very much for your encouragement. Please find our revised manuscript, as well as a series of point-by-point responses to the reviewer comments. We note that in this revised manuscript, we have fully addressed all comments. We’d like to thank the reviewers for their constructive comments and thoughtful evaluation of our work.
Question 1. Table S1 and Table 2 – there is a lack of that information.
Answer: Thanks for your careful check. We revised the Table information for the article.
Question 2. The manuscript could be also improved by one Figure which could show obtained results in an easier way.
Answer: Thanks very much for your good suggestion to improve our manuscript. We designed Figure 6 to describe the application of biochar NPs as nanocarrier in Chinese cabbage growth.
Figure 6. The potential mechanism of BNPs working as nanocarrier to regulate the
Chinese cabbage growth.
Question 3. Line 41 -43 – something is missing in that statement…
Answer: Thanks for your good comment. We revised the sentence on Line 41 to 43 in the revised manuscript “Chinese cabbage (Brassica rapa) is named as napa, napa cabbage, petsai, wongbok, chihli in Asian countries; and it is also known as Chinese leaves or celery cabbage which belongs to the Pekinensis group.”.
Question 4. Line 80 – “platform” – replace word..
Answer: Thanks for your good comment. We switched “platform” to “material” on the Line 80 “In this study, we used biochar nanoparticles (BNPs) as a kind of material for fertilizers to increase Chinese cabbage growth.”
Question 5. There are no aims or hypotheses. The reader should get information about what you did and why, what you expected, and what was your hypothesis. Please provide them…
Answer: Thanks for your good suggestion to improve our manuscript quality. We added the research hypothesis in the revised manuscript.
“In this study, it is hypothesized that biochar nanoparticles (BNPs) are a kind of material that could reduce the fertilizers use for plant growth.”
Question 6. please provide one Figure which presents the experimental design, it is hard to follow your experiment just on the provided description. There are many experimental treatments, therefore it would be ideal to present both Exp. On one scheme, you did Exp. 1 and on the base of the obtained data you did Exp. 2. Please show the relationship between those two experiments.
Answer: Thanks for your good comment. We presented the treatment information for Experiment I and Experiment II in Table S2 and Table S3 in the Supplemental Information, respectively.
Table S2 Experiment I: Response Surface Model for Screening Experiment
Treatments |
N |
P |
K |
T1 |
0 |
0 |
0 |
T2 |
0 |
2 |
2 |
T3 |
1 |
2 |
2 |
T4 |
2 |
0 |
2 |
T5 |
2 |
1 |
2 |
T6 |
2 |
2 |
2 |
T7 |
2 |
3 |
2 |
T8 |
2 |
2 |
3 |
T9 |
2 |
2 |
0 |
T10 |
2 |
2 |
1 |
T11 |
3 |
2 |
2 |
T12 |
1 |
3 |
2 |
T13 |
1 |
1 |
2 |
T14 |
1 |
2 |
1 |
T15 |
2 |
1 |
1 |
Table S3 Experiment II: The Treatments for Experiment of Comparison |
|||
Treatments |
Biochar NPs |
N |
K |
T1 |
1 |
2 |
2 |
T2 |
2 |
2 |
2 |
T3 |
2 |
1 |
1 |
T4 |
2 |
0.5 |
0.5 |
T5 |
2 |
0 |
0 |
T6 |
0 |
2 |
2 |
Question 7. Line 124-128 – should be placed after information from Line 114.
Answer: Thanks for your good suggestion to improve our manuscript quality. We put the Line 124 to 128 after Line 114 in the revised manuscript.
“In the experimental treatments, the BNPs were added to the N-P-K solution in a 1:1 (w/w) ratio. Before the root application, the BNPs and N-P-K solution were mixed together and then placed in a shaker at 140 rpm for 12 hours. In terms of the fertilizer additions, N was derived from calcium nitrate tetrahydrate, P was derived from sodium dihydrogen phosphate dihydrate, and K was derived from potassium sulfate. All the chemicals were of analytical pure grade and were purchased from Nanjing Chemical Reagent Co., Ltd. Leaf quality and stem diameter determine Chinese cabbage grade.”
Question 8. There is no Table S1 (Line 112) , and Supplementary data (Line 111).
Answer: Thanks for your comments. We added the Supplementary Information in the submission. Is that possible to check with the editor? Other reviewers can see our Supplementary Information.
Question 9. Exp 1. And Exp. 2 - Information about units is not clear. Please rewrite them.
Answer: Thanks very much for your suggestion. We revised the treatment information for Experiment I and Experiment 2 description.
Question 10. Table 1 – is missing (Line 131)
Answer: Thanks very much for your check, we put the Table 1 in Supplemental Information as Table S1. Please check our Supplemental Information in the review system.
Question 11. Table 2 – is missing (Line 145, 147)
Answer: Thanks very much for your check, we put the Table 2 in Supplemental Information as Table S2. Please check our Supplemental Information in the review system.
Question 12. Line 151 - Elemental analysis – maybe it is better – Content of elements
Answer: Thanks very much for your good suggestion. We revised the subtitle of 2.3 Content of elements in the revised manuscript.
Question 13. Did you analyze the content of nitrogen, why not?
Answer: Thanks for your good comment. In this study, we want to figure out the best formula for the Chinese cabbage growth in Experiment I (Screening Experiment) and Experiment II (Comparation Experiment), respectively. The elements were not important as the biomass and phenotype; and we selected the common mineral elements in the Chinese cabbage those are components of enzyme systems, structure, and are essential for immunity system function (Kathpalia and Bhatla, 2018). And we think that the selected K, P, Ca, Mg, Na, and S are representative for the plant growth and development.
Reference
Kathpalia R, Bhatla S C. Plant mineral nutrition[M]//Plant physiology, development and metabolism. Springer, Singapore, 2018: 37-81.
Question 14. Please provide information about the level of increase or decrease of analyzed parameters compared to non-fertilized plants (T1). It would be good to indicate that increase in fertilizer use, not always results in higher production of plants’ biomass.
Answer: Thanks very much for your good suggestion to improve our manuscript quality. We added the describe of the Chinese cabbage in control (untreated) in the revised manuscript.
Question 15. Please indicate the amount of N, P, and K (g/L) which resulted in the best plant growth.
Answer: Thanks very much for your good suggestion to improve our manuscript quality. We added the detailed number of the optimum solution of N-P-K was 1.0-0-1.0 g/L (T4, 2-0-2) in the revised manuscript.
Question 16. Similarly, please provide information in Exp. 2 according to observed changes between treatment groups. Please, indicate which treatment group is your control group and how the treatment used impacted on obtained results.
Answer: Thanks very much for your good suggestion to improve our manuscript quality. We added the describe of the Chinese cabbage in control (untreated) in the revised manuscript.
Question 17. Moreover, please use the same names for treatment groups, thus Fig 4 and 5 are not the same as used in the description of Exp. 2, placed in the manuscript part: material and methods. Please, check carefully all the figures description…
Answer: Thanks for your good comments. We revised the names of the treatment groups in Fig. 4 and 5 in the figure notes.
Question 18. There is no clear information, that in exp 2 you did not add P. Please add it.
Answer: Thanks very much for your good suggestion to improve our manuscript quality. We added the reason we select best formula of N-P-K (T4, 2-0-2, 1.0-0-1.0 g/L) for experiment II.
Question 19. Did you check the content of elements in the soil after the experiment ends, so you could point out that Biochar NPs significantly impact the outflow of elements?
Answer: Thanks very much for your good suggestion to improve our manuscript quality. We only want to figure out the best formula for biochar NPs as nanocarrier for fertilizer use. And we will continue the research that how the biochar NPs working as the nanocarrier to regulate the soil quality and improve the elements transfer in the next study.
Question 20. Line – 248 – you did not analyze the effect of phosphate-solubilizing bacteria, so please remove these statements.
Answer: Thanks very much for your careful check for our manuscript. We deleted the description of the phosphate-solubilizing bacteria on Line 248 in the revised manuscript.
Question 21. Line 250 – “potential reason” – potential, not intended?
Answer: Thanks for your good suggestion. We revised the potential reason to the intended reason in the manuscript.

Round 2
Reviewer 4 Report
I have no comments